# Environmental Exposure to Pesticides and Breast Cancer in a Region of Intensive Agribusiness Activity in Brazil: A Case-Control Study

**DOI:** 10.3390/ijerph16203951

**Published:** 2019-10-17

**Authors:** Ageo M. C. Silva, Paulo H. N. Campos, Inês E. Mattos, Shakoor Hajat, Eliana M. Lacerda, Marcelo J. M. Ferreira

**Affiliations:** 1Post Graduated Program of Environment and Health, Cuiaba University, Cuiabá 78065900, Brazil; paulo_campos76@hotmail.com; 2National School of Public Health, Fundação Oswaldo Cruz, Rio de Janeiro 21041-210, Brazil; iemattos@yahoo.com; 3London School of Hygiene & Tropical Medicine, London WC1H 9SH, UK; Shakoor.hajat@lshtm.ac.uk (S.H.); Eliana.Lacerda@lshtm.ac.uk (E.M.L.); 4Post Graduated Program of Public Health, Medicine School, Federal University of Ceará, Fortaleza 60430-140, Brazil

**Keywords:** breast cancer, environmental exposure, pesticides, health surveillance, environmental health

## Abstract

*Background*: Breast cancer is a serious public health problem and is the second most prevalent cancer type in the world. The purpose of this article is to evaluate the association between pesticide use and breast cancer in a region of intense agribusiness activity in the state of Mato Grosso, Brazil. *Methods*: A case-control study was conducted on women living in the city of Rondonópolis, in the south of Mato Grosso state. There were 85 cases of women with confirmed breast cancer and 266 controls who were randomly selected from primary health care users. Bivariate and stratified analyses were performed. Multiple logistic regression was then performed, keeping in the final model the factors with a significance level lower than or equal to 0.05 or considered important according to apriori biological criteria. *Results*: In the final model, living near cropland with pesticides (OR: 2.37; CI: 95% 1.78–3.16) and women aged over 50 years who experienced early menarche (OR: 2.08; CI: 95% 1.06–4.12) had a higher risk of developing breast cancer compared to control subjects. *Conclusion*: This study highlights the importance of exposure to pesticides as an environmental risk factor for the development of breast cancer among women.

## 1. Introduction

Cancer is considered a serious public health problem. Estimates indicate that in 2018, more than 17 million new cases were registered worldwide [1]. In addition, it has a high fatality rate. Globally, one in six deaths are related to the disease. In 2018 alone, it was responsible for more than 9.6 million deaths [2].

Among the types of cancer, breast cancer is the second most prevalent, with approximately 2.09 million cases, second only to lung cancer [1]. It is also the most commonly diagnosed among women, representing a quarter of all recorded cases in the world [3]. It is still considered the leading cause of cancer-related deaths among women [4].

Despite technological advances and the encouragement of health promotion strategies, the prevalence of breast cancer continues to increase. However, it is not evenly distributed in the world. In recent decades, there has been a trend of growing incidence and mortality rates of this type of cancer in developing countries compared to developed countries [1,3].

The pathophysiology of breast cancer is complex and several risk factors are associated with its occurrence. They include elements related to genetic mutations, reproductive health, family history, and occupational and environmental exposures to contaminants [5,6].

Chemical agents are one of the major sources of occupational and environmental exposure that can increase the risk of developing breast cancer. Among the main toxic cancer-related substances are disinfectants, lubricants, chemical solvents, paints, industrial oils, heavy metals, and pesticides [7,8].

Regarding pesticides in particular, several studies have suggested their potential genotoxicity, immunotoxicity, ability to act as tumor promoters and as endocrine disruptors, as the pathophysiology mechanisms related to the increasing risk of breast cancer [9,10,11,12,13,14]. In humans, a case-control study including 1169 cases and 1743 controls was conducted in Canada from 2009 to 2011. During this period, a 43% increase in breast cancer risk (OR: 1.43; 95% CI: 1.15–1.78) in women indirectly exposed to pesticides was seen [15]. Another population-based case-control study identified a high risk of breast cancer development in rural populations, suggesting the importance of collecting in-depth workers’ occupational histories, as well as a detailed assessment of environmental risk factors [14].

In Brazil, it is estimated that there are over 119,000 new cases of breast cancer in women in the 2018–2019 biennium alone. This condition represents almost 30% of reported cases of female cancer in the country. It is also the one that kills most in the female population, accounting for 16.1% of deaths [16]. According to the National Cancer Institute (INCA), of all reported cases, about 60% to 70% are associated to a greater or lesser extent with environmental factors, including exposure to pesticides [17].

In the national context, this data becomes relevant, considering that since 2008 Brazil has been leading the world ranking as the country that imports the most pesticides in the world, accounting for 86% of the total pesticides traded in Latin America [18]. In 2019 alone, the Brazilian Federal government legalized the use of 166 new chemicals, totaling a number of 2232 pesticides currently in circulation [19].

Besides the high volume of usage and chemical variety, pesticides used in Brazil are among the most toxic. A study analyzed the 20 most commonly used active ingredients from 2012 to 2016. Of those, 15% were classified as extremely toxic, 25% as highly toxic, and 35% as moderately toxic [20]. According to the National Health Surveillance Agency (ANVISA), of the 50 most used pesticides in our country, 22 are banned in the European Union. This makes Brazil the largest consumer of banned pesticides than other countries in the world [21].

Among the states with the highest consumption of pesticides in Brazil, Mato Grosso comes first with about 207 million liters applied to its plantations [20]. Seven of the ten municipalities that most sprayed pesticides in the country in 2015 are located in the state of Mato Grosso [20]. This means that a large part of its population is exposed directly or indirectly to pesticides. The most used active ingredients of pesticides used in soybean, corn, and sugarcane crops are Glyphosate, 2,4 Dichlorophenoxyacetic, and Trifuralin, which have been associated with breast cancer [22,23,24]. In addition, in 2018, breast cancer was the most commonly reported, representing 48.98% of all registered cases of cancer among women in Brazil [25]. Given this context, the present study aimed to evaluate the association between pesticide exposure and breast cancer in a region of intense agribusiness activity in the state of Mato Grosso.

## 2. Materials and Methods

### 2.1. Study Type and Location

This is a case-control study on women living in the city of Rondonópolis, located in the south of Mato Grosso state. Data from the Brazilian Institute of Geography and Statistics (IBGE) estimated a local population of 228,857 people in 2018. The main crops in the region are soybeans and corn. Together, these crops have a production of over 450,000 tons of grain. In terms of area, it covers a total of 115,000 hectares [26]. In 2018, the municipality was the main grain exporter of Mato Grosso, totaling US$ 1.318 billion. This value is equivalent to 12.8% of the state’s exports.

### 2.2. Sample Calculation and Inclusion and Exclusion Criteria

The estimated sample size was 351 women (85 cases and 266 controls) which was enough to detect with a significant level of 95% and 5% sampling error with a ratio of controls:cases of 3:1. The maximum expected exposure frequency of the controls was 40% with a minimum odds ratio of 2.1 with a significant level of 95% and power of association of 80%. Cases and control were recruited from November 2017 to June 2018. As cases, we considered the women recruited from the Specialized Oncology Therapy Centre in the city of Rondonópolis-MT (NUTEC), who had breast cancer and whose confirmation was made through histopathological examination. Controls were randomly selected among users of primary health care (PHC) in the city of Rondonópolis, which as of 2019 has 37 basic health units (UBS) with a 77% coverage for primary health care [27]. For this, we used the case hospitalization forms as an aid to the choice of controls in the territory, so that it was possible to establish a closer correlation between cases and controls. Lastly, units were chosen by simple random sampling to select a total of 24.3% of the UBS, totaling 9 UBS. The first UBS chosen was the one located at the most central area of the municipality. The others were selected according to the cardinal points (two for each point), starting from the most peripheral and finishing at the epicenter.

Additional inclusion criteria were residency in the city for at least five years and a minimum age of 20 years old or older at the date of the interview. Women with a previous history of any cancer were excluded.

### 2.3. Instrument and Variables Collected

The interviews were conducted in order of arrival of the consenting women participants. A structured questionnaire was designed and applied by duly trained field researchers, in order to guarantee the standardization of the procedures performed for data collection. The questionnaire was divided into five blocks:
Sociodemographic characteristics: age (≤50 years/>50 years); self-reported race/color (white/other); marital status (married and stable union/others); education (secondary school, incomplete/secondary school, completed/6th form college, incomplete/6th form college, completed/graduate/post-graduate); family income (over 1200.00 reais/up to 1200.00 reais).Life habits: past alcohol consumption (Yes/No); frequency and length of use, including type of drink; past tobacco use (Yes/No), including period of time smoking and quantity of cigarettes consumed.Individual, clinical and hereditary variables: weight; height; BMI (18.5 ≤ 25; >25) age of menarche (9–12 years/>13 years); previous abortion (Yes/No); premature children; breastfeeding (Yes/No); duration of breastfeeding (did not breastfeed or breastfed less than 1 year/breastfed more than 1 year); contraceptive use (Yes/No); menopause age (self-reported); hormone replacement therapy for more than five years (Yes/No); performance of mammograms.Environmental exposure assessment: history of rural or urban residence; residence history of up to 500 m from crops using pesticides; history of pesticide use at home; history of pesticide use and food consumption from vegetable patch, orchard or garden; history of residence near mines or factories.Occupational history: work history in exposed activities or direct contact with pesticides; types of pesticides; history of use of personal protective equipment (PPE) in the handling of pesticides; history of laundering clothes of a person who worked with pesticides; personal or partner’s work history in a rural zone.


### 2.4. Data Analysis

We carried out a descriptive analysis of the variables, followed by a bivariate analysis, where the crude associations between exposure and effect were identified. Stratified analysis was performed using the Mantel–Haenzel chi-square test to select possible confounding or interaction variables, according to the biological plausibility criteria endorsed by the scientific literature. Finally, a logistic regression was performed considering all variables that in the crude and stratified analyses presented the *p*-value ≤ 0.10. In the final model, those with a significance level less than or equal to 0.05 or those considered a priori, according to the literature, remained. Analyses were performed using Epi-Info^®^ 7 software (Center for Desease Control and Prevention, Atlanta, Georgia, USA), with subsequent use of SPSS^®^ version 18.0 software (SPSS Inc., Chicago, IL, USA) for multiple analysis. The research project was approved by the Research Ethics Committee of the University of Cuiabá under number 2.358.088. All participants were informed about the study features and signed the Informed Consent Form.

## 3. Results

The distribution of sociodemographic variables is shown in Table 1. The mean age was 54.3 years for the cases (±8.4 years) and 51.1 years for the controls (±10.6) (results not described in table). More than half of the cases and controls had a level of education up to secondary school/6th form college, with a family income of over R$ 1000 (Table 1).

Table 2 presents the results of the bivariate analysis. To belong to the age group of up to 50 years (OR: 1.78; 95% CI: 1.05–3.00), to be married or to be in a stable union (OR: 1.97; 95% CI: 1.02–2.70), to have a family income of over R$ 1200/month and to report breastfeeding time of less than 1 year or no breastfeeding at all (OR: 1.76; 95% CI: 1.05–2.93) were statistically associated with breast cancer (Table 2).

Regarding the environmental and occupational exposure variables, living close to crops that applied pesticides was significantly associated with the occurrence of breast cancer (OR: 2.13; 95% CI: 1.09–4.16) (Table 3).

In the stratified analysis by age group, it was observed that in the stratum of women aged 50 and over, the report of menarche between 9 and 12 years showed a statistically significant association with the disease (OR: 1.96; 95% CI: 1, 05–3.65). In the stratum of women under 50 years, this association was not observed (Table 4).

In the final logistic regression model, being married or being in a stable union presented as a borderline variable. However, living close to cultivated areas using pesticides (OR: 2.37; 95% CI: 1.78–3.16) and the interaction time between the age group above 50 years and early menarche (OR: 2.08; 95% CI: 1.06–4.12) remained statistically associated with the outcome (Table 5).

## 4. Discussion

In the final analytical model after adjustments, living near plantation areas with pesticide application increased the odds of breast cancer among women by 2.37 times. This result is of great relevance, mainly because it is in line with other studies that show that human exposure to environmental pollutants, especially pesticides, is associated with the growing tendency of several types of cancer, including breast cancer in women [7,9,18,28,29].

Exposure to pesticides in regions of medium and high agricultural production has been shown to be a potential cause of cancer, especially in homes near areas of pesticide use [30]. A case-control study conducted in several regions of Spain identified an association between breast cancer and living near agricultural areas with exposure to pesticides [12]. In Pakistan, research has evaluated the presence of pesticides in dust and their correlation with blood and urine metabolites of potentially agriculturally exposed individuals. The results indicated a greater presence of these biomarkers in exposed individuals when compared to the control group [31].

The state of Mato Grosso is the largest consumer of pesticides in Brazil, accounting for 18.9% of the total consumed in the country. In addition, among the fifty most commonly used active ingredients in their crops, twenty have epidemiological evidence that they act as mutagenic and teratogenic agents [32]. As a consequence, several studies have shown human and environmental contamination in the region. Among them, the case of the extended rural accident resulting from aerial spraying of pesticides that reached the urban space of Lucas do Rio Verde [33] stands out. Another study identified the presence of pesticide residues even in rainwater, in addition to aerial spraying in schools. Thus, it is clear that exposure to pesticides transcends the borders of agricultural areas, reaching rural workers, urban dwellers, children, young people, and adults [18].

The relationship between environmental exposure to pesticides and breast cancer evidenced in our study is also worrying from the point of view of public policies. Our findings reinforce a public health warning that clashes with the policy of liberalization of active pesticide ingredients carried out by the current federal government. In this sense, it strengthens the need to rethink the foundations of the chemical-dependent production model in order to protect human life and the environment.

Also, in the final model, women 50 years of age and older who reported early menarche were 2.08 times more likely to develop breast cancer. This finding is significant, especially due to two aspects: First, it shows that the increasing trend in cancer incidence cannot be attributed only to the ageing of the population, but also due to the diffusion of carcinogenic agents in the environment [22]; second, it reinforces the hypothesis that chronic exposure to low doses of pesticides in different life cycles of human development produces a general increase in carcinogenic processes [22,34]. In the same vein, a meta-analysis study including 118,964 women with invasive breast cancer and 306,091 without the disease found that the risk for the disease increased by 5% for each year of early onset menarche [35]. Another study at the Sichuan Cancer Hospital in China found that early menarche was associated with an increased risk of breast cancer compared to women who started menarche at a later age [36].

In the bivariate analysis, not breastfeeding or breastfeeding for less than one year showed a statistical association with the onset of breast cancer. This correlation is important because it reinforces the hypothesis that breastfeeding may also contribute as a protective factor for the disease. In this vein, a retrospective study analyzed 504 anamneses of patients diagnosed with breast cancer in a referral hospital in Granada, Spain. Among the conclusions, it was evidenced that breastfeeding for more than 1 year acts as a protective factor against breast cancer [37]. A systematic review that included 8 cohort studies and 19 case-control studies covering 36,881 breast cancer cases also showed the protective effect of breastfeeding on breast cancer [38]. A meta-analysis research that included 27 cohort and case-control studies found that prolonged breastfeeding duration was inversely associated with the risk of breast cancer [39].

In this study, it was observed that women who were married or in a stable union with higher family income had higher risk of breast cancer. Jemal et al., report that transitions in the Human Development Index (HDI) lead to changes in the scale and profile of cancer occurrence. In addition to breast cancer, several other cancers have been diagnosed more frequently in areas with higher HDIs. It is possible that the survival of women with cancer is higher among those in better financial conditions [40]. These differences may be related to the various modes of access, detection, and treatment for the disease among the more or less developed regions of Brazil [34].

Still in the bivariate analysis, it is important to observe the increase of the OR as exposure to pesticides occurs more directly in the study population. Working in rural zones, laundering clothes of a person who worked with pesticides, and working with pesticides increased the risk of developing breast cancer by 55%, 80%, and 96%, respectively. Although confidence intervals do not suggest a statistical correlation with increased risk for breast cancer, they may serve as warning indicators for the development of new studies on the effects of environmental pesticide contamination and its impacts on human health.

### Study Limitations

Some limitations on the methods should be considered, including the possibility of recall bias inherent in case-control studies. Although these data were accurately collected, it is necessary to consider the potential bias of the informants, which is characteristic of case-control studies. However, this study sought to minimize this bias through intensive training of field researchers, to enhance rigor in the interview process of research participants and minimize bias.

## 5. Conclusions

This study highlights the importance of different environmental exposures as risk factors for the occurrence of breast cancer. These findings are important because they reinforce the need to consider exposure to pesticides beyond the occupational dimension. Environmental monitoring should be prioritized, especially in areas that have large territorial extensions intended for agribusiness cultivation.

In addition, it reinforces the role and performance of health surveillance in general, and environmental and health surveillance of workers in particular, considering their responsibilities in proposing actions to identify and monitor potential pathways of exposure to environmental contaminants in the population. These processes are important, especially in states such as Mato Grosso, where a large part of the population is directly or indirectly exposed to pesticides. Thus, it is expected that these results help in the prevention, diagnosis, and early treatment of women with breast cancer not only in the researched region, but also throughout Brazil.

Finally, this study also problematizes the current federal government’s policy choice of releasing more than two hundred active pesticide ingredients for commercialization in Brazil in 2019. Such measures further aggravate the chances of environmental and human contamination from exposure to pesticides and may result in the emergence of various pathologies such as breast cancer.

## Figures and Tables

**Table 1 ijerph-16-03951-t001:** Distribution of socio-demographic characteristics of cases and controls associated with breast cancer, Rondonópolis-MT, November 2017 to April 2018.

Variable	Total		Cases		Controls	
	*n*	%	*n*	%	*n*	%
Race/color						
East Asian	1	0.28	1	1.18	-	-
Black	65	18.52	14	16.47	51	19.17
White	107	31.05	20	23.53	89	33.46
Brown	176	50.14	50	58.82	126	47.37
Marital status						
Single	61	17.38	12	14.12	49	18.42
Married/stable union	198	55.55	50	58.82	145	54.51
Widowed	44	12.54	11	12.94	33	12.41
Divorced	48	13.68	12	14.12	36	13.53
No information	3	0.85	-	-	3	1.13
Level of education						
Did not complete secondary school	120	36.36	21	26.92	99	39.29
Completed secondary school/did not complete 6th form college	93	28.18	31	39.74	62	24.60
Completed 6th form college/did not complete university	79	23.94	19	24.36	60	23.81
Completed university/did not complete post-graduate course	38	11.52	7	8.97	31	12.30
Family income						
~l R$ 500.0	3	0.86	-	-	3	1.14
R$ 501 1–R$ 800.0/month	6	1.72	-	-	6	1.52
R$ 801 l–R$ 900.0/month	37	10.60	5	5.88	32	12.12
R$ 901 l–R$ 1000.0/month	48	13.75	11	12.94	37	14.02
R$ 1000.0 l~/month	255	73.07	69	81.18	186	70.45

**Table 2 ijerph-16-03951-t002:** Bivariate analysis of socio-demographic factors and breast cancer, Rondonópolis-MT, November 2017 to April 2018.

Variable	Case		Control		*OR*	*95% CI*		*p* Value
	*n*	%	*n*	%		Lower Limit	Upper Limit	
Age group								
Up to 50 years old	59	69.41	149	56.02	1.78	1.05	3.00	0.02
50 years old and above	26	30.59	117	43.98	1.00			
Race/color								
White	20	23.53	89	33.46	1.63	0.93	2.86	0.08
Others	65	76.47	177	66.54	1.00	-	-	-
Marital status								
Casada/stable union	41	48.24	95	35.71	1.67	1.02	2.7	0.03
Others	44	51.76	171	64.29	1.00	-	-	-
Level of education								
Incomplete secondary school	21	26.92	99	39.29	0.93	0.36	2.48	0.89
Completed secondary school/6th form incomplete	31	39.74	62	24.60	2.21	0.87	5.59	0.09
Completed 6th form/incomplete university	19	24.36	60	23.81	1.40	0.53	3.69	0.49
Completed/post-grad	7	8.97	31	12.30	1.00			
Family income								
Over R$ 1.200/month	69	81.18	186	70.45	1.80	1.01	3.39	0.05
Up to R$ 1.200/month	16	18.82	78	29.55	1.00	-	-	-
Past alcohol consumption								
Yes	39	45.88	123	46.24	0.98	0.60	1.60	0.95
No	46	54.12	143	53.76	1.00	-	-	-
Past smoker								
Yes	18	21.18	74	26.21	0.69	0.38	1.25	0.22
No	67	78.82	192	72.18	1.00	-	-	-
BMI								
Over 25	56	67.47	154	59.92	1.38	0.82	2.33	0.21
<18.5–25	27	32.53	103	40.08	1.00	-	-	-
Menarch								
From 9 to 12 years old	51	60	132	50	1.50	0.91	2.46	0.10
13 years old and above	34	40	132	50	1.00	-	-	-
Breastfeeding								
No	4	5.00	8	3.15	1.61	0.41	5.49	0.44
Yes	76	95.00	246	9685	1.00	-	-	-
Amount of time breastfeeding								
Did not breastfeed/breastfed for less than 1 year	40	50.63	93	36.76	1.76		2.93	0.02
Breastfed more than 1 year	39	49.37	160	63.24	1.00	-	-	-
Use of contraceptive								
Yes	67	78.82	194	73.93	1.38	0.76	2.48	0.27
No	18	21.18	72	27.07	1.00	-	-	-
Hormone Replacement Therapy for over 5 years								
Yes	7	8.24	24	9.33	0.89	0.37	2.15	0.80
No	78	91.76	239	90.87	1.00	-	-	-

**Table 3 ijerph-16-03951-t003:** Analysis of occupational and environmental exposure to pesticides and breast cancer, Rondonópolis-MT, November 2017 to April 2018.

Variable	Case		Control		*OR*	CI		*p* Value
	*n*	%	*n*	%		Lower Limit	Upper Limit	
Pesticides used at home								
Yes	41	48.24	123	46.24	1.08	0.66	1.76	0.74
No	44	23.53	143	53.76	1.00	-	-	-
Pesticides used in vegetable patch, orchard or garden								
Yes	45	52.33	81	44.91	1.34	0.83	2.19	0.23
No	41	47.67	185	55.09	1.00	-	-	-
Residence near a crop field using pesticides								
Yes	73	85.88	197	74.06	2.13	1.09	4.16	0.02
No	12	14.12	69	25.94	1.00	-	-	-
Worked with pesticides								
Yes	6	7.79	10	4.12	1.96	0.69	5.60	0.19
No	71	92.21	233	95.88	1.00	-	-	-
Laundered clothes of a person who worked with pesticides								
Yes	6	7.32	10	4.12	1.80	0.63	5.11	0.25
No	76	92.68	228	95.88	1.00	-	-	-
Worked in a rural zone								
Yes	53	62.35	133	51.55	1.55	0.94	2.57	0.08
No	32	37.65	125	48.45	1.00	-	-	-
Had partner who worked in a rural zone								
Yes	25	29.76	78	30.23	0.97	0.57	1.67	0.93
No	59	70.24	180	69.77	1.00	-	-	-

**Table 4 ijerph-16-03951-t004:** Stratified analysis by age and breast cancer, Rondonópolis-MT, November 2017 to April 2018.

Variables	Cases		Controls		OR	CI		*p* Value
	*n*	%	*n*	%		Lower Limit	Upper Limit	
Age range up to 49 years old								
Menarch from 9 to 12 years old	13	50	61	52.59	0.90	0.38	2.11	0.81
Menarch 13 years old and above	13	19.12	55	80.88	1.00	-	-	-
Age range 50 years old and above								
Menarch from 9 to 12 years old	38	64.41	71	47.97	1.96	1.05	3.65	0.03
Menarch 13 years old and above	21	35.59	77	78.57	1.00	-	-	-

**Table 5 ijerph-16-03951-t005:** Final analytical model for the occurrence of breast cancer, Rondonópolis-MT, November 2017 to April 2018.

Variables	OR Crude	CI		*p* Value	OR Adjusted	CI		*p* Value
		Lowe Limit	Upper Limit			Lowe Limit	Upper Limit	
Age group *	1.78	1.05	3.00	0.02	-	-	-	-
Race/Skin color	1.63	0.93	2.86	0.08	-	-	-	-
Family income	1.80	1.01	3.39	0.05	-	-	-	-
Menarch from 9 to 12 years old	1.50	0.91	2.46	0.10	-	-	-	-
Did not breastfeed/breastfed for less than 1 year	1.76	1.05	2.93	0.02	-	-	-	-
Age group * Menarch	-	-	-	-	2.08	1.06	4.12	0.03
Residence near a crop field using pesticides	2.13	1.09	4.16	0.02	2.37	1.78	3.16	0.02
Married/stable union	1.67	1.02	2.7	0.03	1.96	1.02	2.79	0.05
Worked with pesticides	1.96	0.69	5.60	0.19	-	-	-	-
Worked in a rural zone	1.55	0.94	2.57	0.08	-	-	-	-

* Adjustment variable.

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
