# Peer review of "Environmental Exposure to Pesticides and Breast Cancer in a Region of Intensive Agribusiness Activity in Brazil: A Case-Control Study"

_ijerph, 2019, doi:10.3390/ijerph16203951_

Round 1

Reviewer 1 Report

(1) The major weakness of this study is the choice of 266 controls who were randomly selected among users of Primary Health 99 Care (PHC) in the city of Rondonópolis, which as of 2019 has 37 Basic Health Units (UBS). I assume that the results will be different if you use other different 266 persons as controls. The best controls will be the whole UBS population from November 2017 to June 2018 (the same time as cases). The question is "how do you know these 266 controls are a good representatives?"

(2) The sample size justification is not complete. The manuscript states "351 women (85 cases and 266 controls) with was enough to detect Odds Ratio with a significant level of 95% and power of 80%, with a ratio of controls:cases of 3:1". How big odds ratio (OR) can be detected using this sample size?  It was 2 or 3? I am guessing 266 controls may not be good enough to represent the "real" controls. 

(3) It is more common to say "univariate analysis", not bivariate analysis; as a contrast, it is better to say "a multiple logistic regression model".

(4) In Data Analysis section, I understand you use logistic model to get p-value, but what are the tests to produce p-values in stratified analysis and/or bivariate analysis, respectively? Such as chi-square test or Mantel-Haenszel test?

(5) Table 2: 95%IC should be 95%CI; in Table 3, 4 and 5: IC should be 95%CI.    

Author Response

REVIEW 1 -

The major weakness of this study is the choice of 266 controls who were randomly selected among users of Primary Health Care (PHC) in the city of Rondonópolis, which as of 2019 has 37 Basic Health Units (UBS). I assume that the results will be different if you use other different 266 persons as controls. The best controls will be the whole UBS population from November 2017 to June 2018 (the same time as cases). The question is "how do you know these 266 controls are a good representatives?"

We thank you the reviewer for pointing this issue. We are aware that that one of the main problems in case-control studies are related to the control group(s), particularly, how they would well represent the population where the cases were drawn. This is why we used the cases’ health records to identify addresses and to recruit the controls from the primary care clinics (UBS) from where the cases were registered. We also agree with the reviewer when speaking of the date of recruitment of study controls. Both happened over the same period of time and we made this clearer in the Material and Methods section. Another important point that was added in the methods is the coverage of Primary Health Care in that municipality, which exceeds 77%, which puts us in a comfortable position regarding their choice and representativeness Thus, they were likely to have similar environmental exposures, as they live in the same UBS catchment areas as the cases. We agree that ideally, we should recruit the entire population of non-cases, but this is hardly the case in epidemiological studies, particularly due to time and costs constrains.  We rely on statistical methods to calculate a sample size with parameters that enable us to make assumptions with enough power and a significant level of 95%.  Thus, the representativeness of the control group, in statistical terms is described in the Materials and Methods section.

The sample size justification is not complete. The manuscript states "351 women (85 cases and 266 controls) with was enough to detect Odds Ratio with a significant level of 95% and power of 80%, with a ratio of controls: cases of 3:1". How big odds ratio (OR) can be detected using this sample size? It was 2 or 3? I am guessing 266 controls may not be good enough to represent the "real" controls.

The authors agree with the reviewer's observation regarding the incompleteness of the data presented. The following information was included in the article: to estimate the sample size, a sampling error of 5% was assumed. The relationship between cases and controls followed the 1:3 distribution. The maximum expected exposure frequency of the controls was 40% with a minimum odds ratio of 2.1 with a significant level of 95% and power of 80%.

It is more common to say "univariate analysis", not bivariate analysis; as a contrast, it is better to say "a multiple logistic regression model".

The requested correction was made from the manuscript. The term "bivatiate" has been replaced by "univariate analysis".

In Data Analysis section, I understand you use logistic model to get p-value, but what are the tests to produce p-values in stratified analysis and/or bivariate analysis, respectively? Such as chi-square test or Mantel-Haenszel test

The authors agree with the reviewer's suggestion. The article supplemented the information on the tests used Mantel-Haenzel

Table 2: 95%IC should be 95%CI; in Table 3, 4 and 5: IC should be 95%CI.

95% CI was replaced to 95% CI.

Reviewer 2 Report

The manuscript is interesting and it is written in easy way. I think that the topic is of interest and even if the research does not deal with innovative topics, it certainly adds some elements of interest to a widely discussed and debated matter as the association between pesticides exposure and breast cancer. However, there are several major concerns on the manuscript before publication:

The mean age was 54.3 years 146 for the cases (± 8.4 years) and 51.1 years for the controls (± 10.6) (results not described in table). The age was not unbalanced between the case and control groups, which should be explained and adjusted in the subsequent analysis. the introduction section was not organized logically, and should be improved. Materials and Methods. My major concern is that the pesticides exposure assessment was based on the self-reported investigation on “History of rural or urban residence; Residence history of up to 500 meters from crops using pesticides; History of pesticide use at home; History of pesticide use and food consumption from vegetable patch, orchard or garden; History of residence near mines or factories. ” etc. The assessment may be not accurate. In addition, there are hundreds of kinds of pesticides. Which kind of pesticides would contribute to the risk of breast cancer? We can not identify the exact information from this study. What is “IC” and “95%IC”? They should be “CI” and “95%CI”, right? Therefore, Line 57, Table 2-5, “IC” and “95%IC” should be changed to “CI” and “95%CI”.

Author Response

The mean age was 54.3 years 146 for the cases (± 8.4 years) and 51.1 years for the controls (± 10.6) (results not described in table). The age was not unbalanced between the case and control groups, which should be explained and adjusted in the subsequent analysis.

In fact, there were statistical differences between the age of the cases and controls, and we maintained this variable in the final logistic regression to account for the potential confounding effect of age. We included this information as a note at the end of table 5.  

The introduction section was not organized logically and should be improved.

We sought to improve the Introduction of the manuscript, by editing a few paragraphs and providing additional information.

Materials and Methods. My major concern is that the pesticides exposure assessment was based on the self-reported investigation on “History of rural or urban residence; Residence history of up to 500 meters from crops using pesticides; History of pesticide use at home; History of pesticide use and food consumption from vegetable patch, orchard or garden; History of residence near mines or factories. ” etc. The assessment may be not accurate. In addition, there are hundreds of kinds of pesticides.

We agree with the reviewer, however there challenges to get good data on pesticides usage is quite common, even in countries that have better regulations in place, which is also true for most environmental risks. We recognized this as limitations of the study. We also argue that while striving to implement regulation and surveillance on pesticides usage, to acquire more reliable data, we should not be prevented to publish findings that are reported by individuals. These data are collected with rigour, and even considering potential informant bias (also now included in the study limitations), is consistent with other findings and can contribute to amounting body of evidence, which at least indicates the need for further investigations.

Which kind of pesticides would contribute to the risk of breast cancer?

Regarding the classes of pesticides, the authors included references, as well as evidence of some of the main active ingredients described in the literature and their relationship with the increased risk of developing breast cancer. In this sense, the inclusion of information can be identified in the text: Several studies point to an association between insecticides and breast cancer. Among them, 2.4 D, Atrazine, Triphaline and Glyphosate stand out. All of these pesticides are widely used in soybean, corn and sugarcane crops throughout the state of Mato Grosso.

We can not identify the exact information from this study. What is “IC” and “95%IC”? They should be “CI” and “95%CI”, right? Therefore, Line 57, Table 2-5, “IC” and “95%IC” should be changed to “CI” and “95%CI”.

Correct corrections were made in each of the tables presented, substituting 95% CI for 95% CI.

Reviewer 3 Report

Case study by Silva et al. show pesticide as an environmental risk factor for the development of breast cancer among women. Report is well written. Below are my comments.

1.      Please elaborate the result section for better understanding.

2.      Please provide the class of pesticides which were associated with high risk for causing breast cancer.

3.      Please also discuss the known role of pesticides in the development of other cancers as well?

4.      What was the rationale for conducting the study linking pesticide usage and the development of breast cancer?

5.      Is there a report estimating the concentration of pesticides in the blood of women who were presented with breast cancer?

Author Response

Please elaborate the result section for better understanding.

The authors worked on reorganization the results section

Please provide the class of pesticides which were associated with high risk for causing breast cancer.

Regarding the classes of pesticides, the authors included references, as well as evidence of some of the main active ingredients described in the literature and their relationship with the increased risk of developing breast cancer.

Please also discuss the known role of pesticides in the development of other cancers as well? 

The field of knowledge about pesticides and their impacts on the development of other cancers is very broad. Depending on the pesticide class, exposure routes and biological plausibility, there is a wide range of possible cancers; thus, it will be out of scope of this study, which was designed specifically to look at breast cancer.

What was the rationale for conducting the study linking pesticide usage and the development of breast cancer?

The reviewer's question is quite relevant. Initially, as shown in the Introduction and Discussion sections, the state of Mato Grosso has the largest volume of pesticides applied to crops. It is also the state with the highest application volume of insecticides such as Glyphosate, 2,4D and Trifuraline, where the literature shows a strong association of these active ingredients with the highest risk for the development of breast cancer. Finally, since 2009, Brazil has been the world champion in pesticide consumption. In addition, the current federal government in less than six months has released more than 200 new active ingredients, a record in the country's history. These factors prompted the investigation reported in our manuscript, which looked at the potential association between environmental exposure to pesticides and breast cancer.

Is there a report estimating the concentration of pesticides in the blood of women who were presented with breast cancer?

We did not have resources to measure traces of pesticides in cases and controls in n this study.

Round 2

Reviewer 2 Report

The authors answered the study design in the cover letter, but the study limitation should be also stated carefully in the text, not just as "...the possibility of recall bias inherent in case-control studies.....such as information related to environmental exposure assessment".

The general information on categories of pesticides used in the area can be listed in a table.

Many errors or confusions on "IC", "CI" and "95%"... have still not been corrected,  for example, why adding lines under CI, 95%CI in table 2, table 3, table 4? Why CI  was used in table 3, table 4, but 95%CI in Table 2? And IC in table 5?

Author Response

REVIEW -

The authors answered the study design in the cover letter, but the study limitation should be also stated carefully in the text, not just as "...the possibility of recall bias inherent in case-control studies.....such as information related to environmental exposure assessment".

We agreed with the reviewer's suggestion and improved the wording of the study limitations.

Many errors or confusions on "IC", "CI" and "95%"... have still not been corrected,  for example, why adding lines under CI, 95%CI in table 2, table 3, table 4? Why CI  was used in table 3, table 4, but 95%CI in Table 2? And IC in table 5?

We agree with the reviewer's suggestions. In fact, it was a translation error that has already been adjusted.

The general information on categories of pesticides used in the area can be listed in a table. 

We have inserted references in the Introduction and Methods sections with the respective categories of pesticides used in the area.
